# Tailored Water-Soluble Covalent Organic Cages for Encapsulation of Pyrene and Information Encryption

**DOI:** 10.3390/ijms242417541

**Published:** 2023-12-16

**Authors:** Haixin Song, Yujing Guo, Guorui Zhang, Linlin Shi

**Affiliations:** College of Chemistry, Zhengzhou University, Zhengzhou 450001, China; 13333771121@163.com (H.S.); yujingguo@zzu.edu.cn (Y.G.)

**Keywords:** covalent organic cage, pyridine salts, water-soluble, cavity structure, encapsulation, information encryption

## Abstract

Forming pyridine salts to construct covalent organic cages is an effective strategy for constructing covalent cage compounds. Covalent organic cages based on pyridine salt structures are prone to form water-soluble supramolecular compounds. Herein, we designed and synthesized a triangular prism-shaped hexagonal cage with a larger cavity and relatively flexible conformation. The supramolecular cage structure was also applied to the encapsulation of pyrene and information encryption.

## 1. Introduction

As an emergent discipline, supramolecular chemistry has been widely known and extensively studied on account of its unique properties and potential applications in biological imitation [1,2,3,4,5,6,7,8], gas encapsulation [9], organic photoreactions [10,11,12], catalysis [13,14], molecular recognition [15,16,17,18], and so on. Many metal-organic supramolecular cages have been constructed, such as octahedrons [19], tetrahedrons [20], spheres [21], square prisms [22], triangular prisms [23], spirals [24], capsules [25], etc.

In addition, covalent organic cages with well-defined intrinsic porosity have attracted increasing attention for the last decade [26,27,28]. Their internal cavity and external channels have been applied for selective recognition and separation [29,30,31], catalysis [32,33], sensing [34,35], and so on. Furthermore, compared with the metal-organic cage, the covalent organic cage has better stability and acid and alkali resistance. At present, the construction of water-soluble cage compounds remains challenging.

Forming pyridine salts to construct covalent organic cages is one of the effective strategies for constructing covalent cage compounds [36,37,38,39]. This strategy can make the covalent organic cage have different solubility by changing the discrete anions outside the cavity. For example, the covalent organic cage with PF_6_^−^ anion is soluble in acetonitrile, and the covalent organic cage with Cl^−^ anion is soluble in water. Therefore, this strategy is also effective for constructing water-soluble cage compounds. In this work, a water-soluble covalent organic cage **C** that can interact with PAHs (polycyclic aromatic hydrocarbons) was designed and synthesized. At the same time, the recycling of covalent organic cage **C** was realized.

## 2. Results and Discussion

Covalent organic cage **C** was synthesized in six steps (Figure 1). Commercially available 1-indanone was selected as the starting substrate for the synthetic route. Compound **2** was obtained by cyclotrimerization, substitution reactions, and bromination reactions by reference to the methods in the literature [40]. Compound **4** was prepared from compound **2** and compound **3** through a Suzuki coupling reaction under the catalysis of tetra-triphenylphosphine palladium in a yield of 62%. Thereafter, an S_N_2 reaction between an excess of *p*-xylylene dibromide (compound **5**) and compound **4** in a MeCN/DMF mixture under reflux for 2 days led to the formation of compound **6** after counterion exchange in a yield of 67%. Finally, equimolar amounts of compound **6** and compound 4 in the presence of 0.2 equiv. of tetrabutylammonium iodide (TBAI) as a catalyst were heated under reflux in MeCN/MeOH/CDCl_3_ mixture solvent for 9 days, resulting in the isolation of the crude chloride salt as a yellow solid after precipitating with tetrabutylammonium chloride (TBACl). The crude chloride salt was washed with a mixture of methanol and ether to give a covalent organic cage **C** in a 9.6% yield.

Covalent organic cage **C** was characterized by ^1^H NMR. The Ha and Hb of compound **6** are in different chemical environments, so Ha and Hb appear as two sets of signal peaks with an integral area ratio of 1:1 (Appendix A). However, after the formation of the covalent organic cage **C**, the Ha and Hb signal peaks in compound **6** transform into an Hc signal peak with the same chemical environment, indicating the formation of a highly symmetric structure (Figure 1a). In the ethyl group on compound **6**, after forming covalent organic cage **C**, the presence of the cavity causes part of the proton signals to move towards the high field, and part of the proton signals are shielded, splitting into Hd, Hd’, and He, He’ (Figure 1b). The splitting of the signal fully proves the formation of the cavity structure and the successful preparation of covalent organic cage **C**.

In addition to hydrogen NMR spectroscopy, electrospray mass spectrometry (ESI-MS) is also a powerful means of characterizing covalent organic cages. Figure 2 shows that the ESI-MS of covalent organic cage **C** revealed four sets of peaks with continuous charge states (3^+^, 4^+^, 5^+^, and 6^+)^, which were attributed to the successful departure of Cl^−^ counterions. The experimental isotope patterns for the four peaks were consistent with the theoretical isotope patterns, indicating that the covalent organic cage **C** has been constructed successfully. At the same time, the strongest peaks have been successfully attributed, and their experimental isotopic patterns are also consistent with the theoretical isotopic patterns (Appendix A). Covalent organic cage **C** was also characterized by high-resolution mass spectrometry (HRMS). In the HRMS of covalent organic cage **C**, peaks with *m*/*z* values of 2005.8201 ([M+H]^+^) were observed (Appendix A) and were shown to be consistent with the calculated values.

Several efforts to obtain complete data for the x-ray analysis of single crystals of covalent organic cage **C** were unsuccessful, so a simulated molecular model of covalent organic cage **C** was constructed using Materials Studio 2020 software (Figure 3a,b). The simulated molecular model analysis revealed that the structure of the covalent organic cage **C** is exactly consistent with the expected triangular prismatic architecture, where the truxene units and phenylene unit centers construct the faces and the pillar. Half of the ethyl chain of the benzene ring pointed into the cavity, while the rest pointed outwards. The window in the hollow cavity of the triangular prism structure is 7.5 Å wide and 18.9 Å long, as measured by the Materials Studio software (Figure 3c).

Covalent organic cage **C**, with a pyridine salt structure, exhibits good fluorescence emission (Appendix A). It has been experimentally found that the covalent organic cage **C** can interact with polycyclic aromatic hydrocarbons via *π*-*π* interactions, where pyrene can gradually quench their fluorescence. As shown in Figure 4a, the fluorescence of the covalent organic cage **C** continuously quenches with the continuous addition of pyrene, and its fluorescence intensity shows a gradual decrease.

After the interaction of the covalent organic cage **C** with pyrene, its recovery and recycling were realized. As shown in Figure 5, the covalent organic cage **C** showed a bright yellow fluorescence in methanol solution, to which pyrene was added, and its fluorescence was gradually quenched. Subsequently, ether and deionized water were added to this mixed system, and the covalent organic cage **C** and pyrene were present in different liquid phases due to their different solubilities (covalent organic cage dissolved in water, pyrene dissolved in ether). Thus, recycling of the covalent organic cage **C** was achieved.

Due to the fact that covalent organic cage **C** can gradually quench its fluorescence when interacting with pyrene, we further explored their applications in fluorescent inks and information encryption using this optical property. As shown in Figure 6, the cartoon image drawn on the filter paper by the methanol solution containing **C** exhibits blue fluorescence, attributed to the covalent organic cage **C** having a pyridine salt structure. Interestingly, when the methanol solution containing pyrene continues to paint other parts of the cartoon image except for the mouth and arm with the methanol solution containing pyrene, we obtain a purple fluorescent image, which may be due to the interaction of covalent organic cage **C** with polycyclic aromatic hydrocarbons through Π-Π interactions in which pyrene can gradually quench its fluorescence. It is important to mention that when a cartoon image is drawn on paper with fluorescent ink (methanol solution containing **C**), only the outline of the pattern can be seen under natural light conditions, but under the irradiation of a 365 nm UV lamp, a blue cartoon image can be clearly seen, and then the fluorescent color changes accordingly under the action of the pyrene solution, so it can be used to display different information according to the color of the fluorescent ink and then used to display different information.

## 3. Materials and Methods

### 3.1. Materials

All reagents were purchased from Sigma-Aldrich, Fisher, Across, and Alfa Aesar and were used without further purification. All solvents were dried according to standard procedures, and all of them were degassed under Ar for 30 min before use. All air-sensitive reactions were carried out under an inert Ar atmosphere.

### 3.2. Measurements

Column chromatography was conducted using SiO_2_ (VWR, 40–60 µm, 60 Å), and the separated products were visualized by UV light. NMR spectra data were recorded on a 600 MHz Bruker NMR spectrometer in CD_3_OD and CD_3_CN with TMS as the reference. The UV-vis spectra were recorded on a dual-beam UV-Vis spectrophotometer (TU-1901). Emission spectra in the liquid state were recorded on a Horiba-FluoroMax-4 spectrofluorometer, and a 1 cm quartz cuvette was employed as the vessel for the recording of the fluorescence emission spectrum. ESI-MS was recorded with a Waters Synapt G2-Si mass spectrometer. High-resolution electrospray ionization mass spectrometry (HR-ESI MS) experiments were performed with a Water Q-Tof Micro MS/MS high-resolution mass spectrometer in ESI mode.

### 3.3. Materials Synthesis

Compound **6** was synthesized according to the literature method [36].

#### 3.3.1. Preparation of Compound **6**

Compound **4** (300 mg, 0.4 mmol) and compound **5** (1.6 g, 6.06 mmol) were dissolved in a 1:1 mixture of DMF/MeCN (32 mL) and refluxed under Ar atmosphere. After 48 h, the reaction mixture was cooled down to room temperature and poured into Et_2_O (159 mL). The yellow precipitate was collected by filtration and washed with CH_2_Cl_2_ (95 mL). Subsequently, the yellow solid was dissolved in DMF (32 mL), followed by the addition of NH_4_PF_6_ (0.5 g) and H_2_O (317 mL). The resulting precipitate was filtered, washed with H_2_O (2 × 30 mL), and dried under vacuum to afford pure compound 6 (yield: 62%). m.p.: 219–220 °C. ^1^H NMR (600 MHz, CD_3_CN) *δ* 8.81 (d, *J* = 6.5 Hz, 2H), 8.68 (d, *J* = 8.7 Hz, 1H), 8.47 (d, *J* = 7.0 Hz, 2H), 8.18 (s, 1H), 8.10 (d, *J* = 8.4 Hz, 1H), 7.69–7.34 (m, 4H), 5.76 (s, 2H), 4.65 (s, 2H), 3.14–3.10 (m, 2H), 2.46–2.43 (m, 2H), 0.25 (t, *J* = 7.3 Hz, 6H). ^13^C NMR (151 MHz, CD_3_CN) *δ* 156.1, 154.0, 147.1, 144.0, 143.6, 139.8, 138.1, 133.2, 132.1, 129.9, 129.1, 128.9, 128.8, 128.8, 128.7, 128.3, 127.3, 126.8, 125.5, 124.9, 122.0, 62.8, 57.4, 32.3, 28.8, 7.6.

#### 3.3.2. Preparation of Covalent Organic Cage **C**

A solution of compound **6** (104.9 mg, 0.14 mmol), compound **4** (203.1 mg, 0.14 mmol), and TBAI (10.4 mg, 0.027 mmol) in dry MeCN (101 mL), MeOH (47 mL), and CHCl_3_ (24 mL) mixture solvent was heated under reflux at 90 °C for 9 days. After cooling to room temperature, an excess of TBACl was added to quench the reaction, and the yellow product was collected by filtration (yield: 9.6%). ^1^H NMR (600 MHz, CD_3_OD_SPE) *δ* 9.07 (d, *J* = 6.4 Hz, 2H), 8.47 (d, *J* = 6.7 Hz, 2H), 8.42 (d, *J* = 8.3 Hz, 1H), 8.10 (s, 1H), 8.02 (d, *J* = 8.5 Hz, 1H), 7.77 (s, 2H), 5.83 (s, 2H), 2.93–2.90 (m, 1H), 2.83–2.77 (m, 1H), 2.34–2.30 (m, 1H), 2.08–2.05 (m, 1H), 0.27 (t, *J* = 7.3 Hz, 3H), -0.61 (t, *J* = 7.2 Hz, 3H).

## 4. Conclusions

In summary, a water-soluble covalent organic cage **C** with six Cl^−^ was resoundingly constructed, followed by detailed ^1^H NMR, ESI-MS, and HRMS spectrometry characterization. Due to its unique pyridine salt structure, the covalent organic cage **C** can exhibit better fluorescence emission. It can also interact with pyrene in *π*-*π* interactions, resulting in a gradual quench of fluorescence. Covalent organic cage **C** can be used in information encryption systems and has broad application prospects in green catalysis and biomedicine due to its unique cavity structure and good water solubility.

## Data Availability

Data are contained within the article.

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
