# Peer review of "Tailored Water-Soluble Covalent Organic Cages for Encapsulation of Pyrene and Information Encryption"

_ijms, 2023, doi:10.3390/ijms242417541_

Round 1

Reviewer 1 Report

Comments and Suggestions for Authors

Zhangi, Shi and co-workers report a water-soluble pyridinum-containing cage capable of recognizing pyrene in aqueous solutions. The obtained compound is quite interesting and can add to the library of similar structures. Unfortunately, when it comes to the state-of-the-art, authors refer to only selected papers neglecting other papers in the field (e.g. Org. Chem. Front., 2022, 9, 81-87). In addition, to be precise in definitions, the cited work 28 is rather about macrocycles than cages. In other words, If the authors do wish to cite more widely, they should pay attention to the works of other groups working in the field. It is also advisable for the authors to emphasize that the recognition of PAHs by pyridinium-containing cages is not a new idea and has already been demonstrated for coronene (ref. 27).

In general, the text contains a lot of jargon, spelling and grammatical errors, confusing wording, so it needs further editing.

Several examples:

Page 1, line 21, should be ‘catalysis’ instead of ‘catalytic’

Page 1, line 42, 1-Indanone should be typed in lowercase

Page 2, line 43, I would not call the cyclotrimerization of 1-indanone to truxene a polymerization reaction.

Page 2, line 53, the ethyl group(s) do not cause anything, it is the presence of the cavity that causes signal shifts.

Page 2, line 63-64, should be rather “hydrogen atom or proton signals” than “hydrogen”.

Page 4, line 89, should be “phenylene” instead of “Benzene”

Page 4, line 97, “Covalent organic cage C has a good fluorescence emission due to its pyridine salt structure” The authors suggest that a similar structure without the pyridinium salt would not exhibit fluorescence properties? Further explanation is needed.

Page 4, line 98, “It was found experimentally that the covalent organic cage C can interact with polycyclic aromatic hydrocarbons via π-π interactions”. What experimental method demonstrated this, and what other polycyclic structures apart from pyrene were investigated?

Page 4, line 99, “pyrene can gradually burst their fluorescence”?

Figure 5. It would be nice if the authors showed real photos, not cartoons, to demonstrate the phenomena described. Besides, I am curious how the authors attained a biphasic mixture using water, methanol, and dimethyl ether, and what are the solvent(s) in the upper and bottom phases. Another disturbing thing to me is the easy dissociation of the complex. Did the authors attempt to measure the binding constant?

Regarding Figure 5, the process of information encryption is not complete, because all elements shown are recognizable. What is about decryption of this image?

The materials and measurements sections as well as the syntheses of 6 and C are duplicated in the Supporting Information.

Summarizing, the manuscript is publishable but after addressing the above concerns.

Comments on the Quality of English Language

See above.

Author Response

Response to the comments of Reviewer #1:

Zhangi, Shi and co-workers report a water-soluble pyridinum-containing cage capable of recognizing pyrene in aqueous solutions.The obtained compound is quite interesting and can add to the library of similar structures.Unfortunately, when it comes to the state-of-the-art, authors refer to only selected papers neglecting other papers in the field (e.g. Org. Chem. Front., 2022, 9, 81-87). In addition, to be precise in definitions, the cited work 28 is rather about macrocycles than cages. In other words, If the authors do wish to cite more widely, they should pay attention to the works of other groups working in the field. It is also advisable for the authors to emphasize that the recognition of PAHs by pyridinium-containing cages is not a new idea and has already been demonstrated for coronene (ref. 27).

In general, the text contains a lot of jargon, spelling and grammatical errors, confusing wording, so it needs further editing.

Several examples:

Page 1, line 21, should be ‘catalysis’ instead of ‘catalytic’

Response: We thank the editor for the encouraging comments, according to the constructive comments of reviewer, we have made revisions to the suggested content in the manuscript, and this change has been marked in the text. Please refer to manuscript for detailed information (line 21).

Page 1, line 42, 1-Indanone should be typed in lowercase

Response: We thank the editor for the encouraging comments, according to the constructive comments of reviewer, we have made revisions to the suggested content in the manuscript, and this change has been marked in the text. Please refer to manuscript for detailed information (Page 1, line 42).

Page 2, line 43,I would not call the cyclotrimerization of 1-indanone to truxene a polymerization reaction.

Response: We thank the editor for the encouraging comments, according to the constructive comments of reviewer, we have made revisions to the suggested content in the manuscript, “cyclotrimerization” and this change has been marked in the text. Please refer to manuscript for detailed information (Page 2, line 43).

Page 2, line 63, the ethyl group(s) do not cause anything, it is the presence of the cavity that causes signal shifts.

Response: We thank the editor for the encouraging comments, according to the constructive comments of reviewer, we have made revisions to the suggested content in the manuscript, and this change has been marked in the text. Please refer to manuscript for detailed information (Page 2, line 63-66).

Page 2, line 63-64, should be rather “hydrogen atom or proton signals” than “hydrogen”.

Response: We thank the editor for the encouraging comments, according to the constructive comments of reviewer, we have made revisions to the suggested content in the manuscript, and this change has been marked in the text. Please refer to manuscript for detailed information (Page 2, line 60-63).

Page 4, line 89, should be “phenylene” instead of “Benzene”.

Response: We thank the editor for the encouraging comments, according to the constructive comments of reviewer, we have made revisions to the suggested content in the manuscript, and this change has been marked in the text. Please refer to manuscript for detailed information (Page 4, line 91).

Page 4, line 97, “Covalent organic cage C has a good fluorescence emission due to its pyridine salt structure” The authors suggest that a similar structure without the pyridinium salt would not exhibit fluorescence properties? Further explanation is needed.

Response: We thank the editor for the encouraging comments, according to the constructive comments of reviewer, this sentence is incorrectly stated. Substances are fluorescent for a variety of reasons, not because they have a pyridine salt structure. The corrected sentence should read: “Covalent organic cage C with a pyridine salt structure has good fluorescence emission”.Please refer to manuscript for detailed information (Page 4, line 99).

Page 4, line 98, “It was found experimentally that the covalent organic cage C can interact with polycyclic aromatic hydrocarbons via π-π interactions”. What experimental method demonstrated this, and what other polycyclic structures apart from pyrene were investigated?

Response: We thank the editor for the encouraging comments, according to the constructive comments of reviewer, covalent organic cages can interact with polycyclic aromatic hydrocarbons through π - π interactions, which can be demonstrated through 1H NMR titration[1], 2D 1H-1H ROESY spectra, independent gradient model (IGM)[2], single crystal X-ray diffraction analysis, and DFT calculations[3]. We have also attempted other polycyclic structures, but covalent cage C has the best effect on quenching pyrene. Therefore, this article mainly demonstrates the interaction of supramolecular cage C with pyrene.

  1. Wu, H.; Wang, Y.; Song, B.; Wang, H.J.; Zhou, J.; Sun, Y.; Jones, L.O.; Liu, W.; Zhang, L.; Zhang, X.,et al. A contorted nanographene shelter. Nat Commun 2021, 12, 5191,doi:10.1038/s41467-021-25255-6.
  2. Lai, Y.L.; Su, J.; Wu, L.X.; Luo, D.; Wang, X.Z.; Zhou, X.C.; Zhou, C.W.; Zhou, X.P.; Dan, L. Selective separation of pyrene from mixed polycyclic aromatic hydrocarbons by a hexahedral metal-organic cage.Chin Chem Lett 2024, 12, 108326, doi:1016/j.cclet.2023.108326.
  3. Ibańez, S.;Gusev,G.; Peris, E. Unexpected Influence of Substituents on the Binding Affinities of Polycyclic Aromatic Hydrocarbons with a Tetra-Au(I) Metallorectangle. Organometallics 2020, 39, 22, 4078-4084, doi:10.1021/acs.organomet.0c00639.

Page 4, line 99, “pyrene can gradually burst their fluorescence”?

Response: We thank the editor for the encouraging comments, according to the constructive comments of reviewer, we have made revisions to the suggested content in the manuscript, it should be changed to “quench” and this change has been marked in the text. Please refer to manuscript for detailed information (Page 4, line 102).

Figure 5. It would be nice if the authors showed real photos, not cartoons, to demonstrate the phenomena described. Besides, I am curious how the authors attained a biphasic mixture using water, methanol, and dimethyl ether, and what are the solvent(s) in the upper and bottom phases. Another disturbing thing to me is the easy dissociation of the complex. Did the authors attempt to measure the binding constant?

Response: We thank the editor for the encouraging comments, according to the constructive comments of reviewer, ether and deionized water are added to the mixed system. Water and methanol are able to dissolve well in each other, while water has a lower solubility in ether, so the mixture can be mixed with ether and then layered. And since pyrene has higher solubility in ether and ether is less dense, the pyrene solution dissolved in ether is in the upper layer, while the covalent cage C dissolved in methanol and water is in the lower layer (Figure S10). On the other hand, we measured the binding constant Ka = 2.7084 × 104 M-1 from the NMR titration curve (Figure S9). Please refer to Supporting Information for detailed information.

Regarding Figure 5, the process of information encryption is not complete, because all elements shown are recognizable. What is about decryption of this image?

Response: We thank the editor for the encouraging comments, according to the constructive comments of reviewer, for the experiment of information encryption, when a cartoon image was drawn on the filter paper with a methanol solution containing C, only the outline of this cartoon image could be seen under natural light conditions, and then when irradiated with a 365nm UV lamp, a clear blue cartoon image could be seen, however, when a methanol solution containing pyrene was used to paint some parts of the cartoon image, a very clear change in color could be seen. The main purpose of decrypting this image is to prove that the addition of pyrene can change the fluorescence of the supramolecular cage, which in turn can be used to encrypt information by utilizing the characteristics of the fluorescence change, such as in the condition of natural light is just an ordinary image, however, in the irradiation of 365nm UV lamp, due to the different fluorescence shown, which in turn can be used to find different information.

The materials and measurements sections as well as the syntheses of 6 and C are duplicated in the Supporting Information.

Response: We thank the editor for the encouraging comments, according to the constructive comments of reviewer, we have made revisions to the suggested content in the manuscript, and this change has been marked in the text. Please refer to Supporting Information for detailed information.

Summarizing, the manuscript is publishable but after addressing the above concerns.

Reviewer 2 Report

Comments and Suggestions for Authors

Song et al. studied the water soluble covalent organic cages for encapsulation of pyrene and information encryption. This study would contribute further understanding of covalent organic cages. Therefore, this paper could be recommended for publication after major revision regarding the following comments.

The authors showed energy-minimized molecular of C. However, the calculation details were not described.

The interaction of organic cage C with pyrene were described by means of photoluminescence spectra. For revealing the photophysical properties, the absorption and photoluminescence spectra of 2, 4, 6, and C are required. 

To clarify the importance of the cage structure, it is effective to compare absorption, photoluminescence, and NMR spectra of C and 4 with titration of that with pyrene.

The structure of pyrene is wrong in Figure 5.

‘The fluorescence of the covalent organic cage C underwent a 6-bay burst when the ratio of pyrene reached 288 times’ in conclusion was not explained in the main manuscript. 

Can the molecular ion peak of the complexs of C and pyrene, and 5 and pyrene be detected by MS analysis? 

Author Response

Song et al. studied the water soluble covalent organic cages for encapsulation of pyrene and information encryption. This study would contribute further understanding of covalent organic cages. Therefore, this paper could be recommended for publication after major revision regarding the following comments.

The authors showed energy-minimized molecular of C. However, the calculation details were not described.

Response: We thank the editor for the encouraging comments, according to the constructive comments of reviewer, all the density-functional theory (DFT) computations were performed using the Gaussian 09 Software Package via the semiempirical PM6 method. The geometry optimization was conducted by the Berny method until the maximum force is below 0.00045 Ha/Bohr and the maximum displacement is below 0.0018 Bohr. Please refer to the Supporting Information for detailed information.

The interaction of organic cage C with pyrene were described by means of photoluminescence spectra. For revealing the photophysical properties, the absorption and photoluminescence spectra of 2, 4, 6, and C are required.

Response: We thank the editor for the encouraging comments, according to the constructive comments of reviewer, we have made revisions to the suggested content in the manuscript, and this change has been marked in the text. Please refer to Supporting Information for detailed information (Figure S7).

To clarify the importance of the cage structure, it is effective to compare absorption, photoluminescence, and NMR spectra of C and 4 with titration of that with pyrene.

Response: We thank the editor for the encouraging comments, according to the constructive comments of reviewer, we have made revisions to the suggested content in the manuscript, and this change has been marked in the text. Please refer to Supporting Information for detailed information (Figure S8-Figure S9).

The structure of pyrene is wrong in Figure 5.

Response: We thank the editor for the encouraging comments, according to the constructive comments of reviewer, we have made revisions to the suggested content in the manuscript, and this change has been marked in the text. Please refer to manuscript for detailed information (Figure 5).

‘The fluorescence of the covalent organic cage C underwent a 6-bay burst when the ratio of pyrene reached 288 times’ in conclusion was not explained in the main manuscript.

Response: We thank the editor for the encouraging comments, according to the constructive comments of reviewer, we have made revisions to the suggested content in the manuscript, and this change has been marked in the text. Please refer to manuscript for detailed information (Page 6, line 183-184).

Can the molecular ion peak of the complexs of C and pyrene, and 5 and pyrene be detected by MS analysis?

Response: We thank the editor for the encouraging comments, according to the constructive comments of reviewer, we have attempted mass spectrometry many times and unfortunately have not been successful in obtaining mass spectra due to the instrumentation or the sample, and the synthesis of the supramolecular cage C has a long synthesis step with a low yield, and it may not be possible to obtain more samples to test in a short period of time. However, we performed NMR titrations to demonstrate the interaction between the compound pyrene and the supramolecular cage. Please refer to Supporting Information for detailed information (Figure S8-Figure S9).

Reviewer 3 Report

Comments and Suggestions for Authors

The manuscript by Zhangi, Shi and co-workers describes the synthesis and characterisation of a specific example of a covalent organic pyridine-based cage compound ("compound C"). It is a short-and-sharp and generally well written and easy to follow report on an interesting topic.

I have a few general concerns and a number of more detailed textual comments and suggestions that the authors would need to address in my view.

I think the fact that this study deals with but one very specific compound would need to be made clear and e.g. be taken into account in the title of the paper.

The second part of the title of this MS is misleading in the sense that only a minute hint is revealed of the change in fluorescent properties brought about by exposure of the painted film to pyrene. This is potentially interesting for encryption, but before such a claim can be made (in the title of a paper!) more detail (and more work) is required. There is no experimental information on this aspect at all present in the current version of the MS.

Some, more textual, comments and suggestions:

the language of the MS is somewhat unusual and "verbose" in places and could be tuned down, e.g. "multitudinous" (line 22), "handily" (l 23), "highly consistent" (l 74 and 77), "cute" (l 117)

Line 34, 35. "easily soluble". What does "easily" refer to? Kinetics? Would just "soluble" (without "easily") suffice?

l 37. Where does the code name "cage C" come from? Explanation needed?

l 38 Acronym PAH needs to be defined at first use

l 58 "appears" needs to be replaced with "appear"

l 90 "outwards" iso "outside"

l 108 "As shown in Fig 5...". I do not think a highly schematic illustration can be used as proof for a mechanism. In my conviction, real experimental data are required here!

l 120 "palm" doesn't seem the right word. "arm" better?

Comments on the Quality of English Language

Apart from some  of my comments above, the English of this MS is adequate.

Author Response

The manuscript by Zhangi, Shi and co-workers describes the synthesis and characterisation of a specific example of a covalent organic pyridine-based cage compound ("compound C"). It is a short-and-sharp and generally well written and easy to follow report on an interesting topic.

I have a few general concerns and a number of more detailed textual comments and suggestions that the authors would need to address in my view.

I think the fact that this study deals with but one very specific compound would need to be made clear and e.g. be taken into account in the title of the paper.

The second part of the title of this MS is misleading in the sense that only a minute hint is revealed of the change in fluorescent properties brought about by exposure of the painted film to pyrene. This is potentially interesting for encryption, but before such a claim can be made (in the title of a paper!) more detail (and more work) is required. There is no experimental information on this aspect at all present in the current version of the MS.

Response: We thank the editor for the encouraging comments, according to the constructive comments of reviewer, we added relevant experimental details. Please refer to manuscript for detailed information (Page 5, line 120-134).

Some, more textual, comments and suggestions:

the language of the MS is somewhat unusual and "verbose" in places and could be tuned down, e.g. "multitudinous" (line 22), "handily" (l 23), "highly consistent" (l 74 and 77), "cute" (l 117)

Response: We thank the editor for the encouraging comments, according to the constructive comments of reviewer, we have made revisions to the suggested content in the manuscript, and this change has been marked in the text. Please refer to manuscript for detailed information.

Line 34, 35. "easily soluble". What does "easily" refer to? Kinetics? Would just "soluble" (without "easily") suffice?

Response: We thank the editor for the encouraging comments, according to the constructive comments of reviewer, we have made revisions to the suggested content in the manuscript, and this change has been marked in the text. Please refer to manuscript for detailed information (Line 34, 35).

l 37. Where does the code name "cage C" come from? Explanation needed?

Response: We thank the editor for the encouraging comments, according to the constructive comments of reviewer, the code name "cage C" comes from the first letter of the cage in English.

l 38 Acronym PAH needs to be defined at first use

Response: We thank the editor for the encouraging comments, according to the constructive comments of reviewer, we have made revisions to the suggested content in the manuscript, and this change has been marked in the text. Please refer to manuscript for detailed information (Line 38).

l 58 "appears" needs to be replaced with "appear"

Response: We thank the editor for the encouraging comments, according to the constructive comments of reviewer, we have made revisions to the suggested content in the manuscript, and this change has been marked in the text. Please refer to manuscript for detailed information (Line 59).

l 90 "outwards" iso "outside"

Response: We thank the editor for the encouraging comments, according to the constructive comments of reviewer, we have made revisions to the suggested content in the manuscript, and this change has been marked in the text. Please refer to manuscript for detailed information (Line 93).

l 108 "As shown in Fig 5...". I do not think a highly schematic illustration can be used as proof for a mechanism. In my conviction, real experimental data are required here!

Response: We thank the editor for the encouraging comments, according to the constructive comments of reviewer, we included experiments to achieve recovery and in-cycling after the interaction of covalent cage C with pyrene, as shown in Figure S10, which demonstrates the change of fluorescence of each part under the irradiation of 365 nm UV lamp.Please refer to Supporting Information for detailed information (Figure S10).

l 120 "palm" doesn't seem the right word. "arm" better?

Response: We thank the editor for the encouraging comments, according to the constructive comments of reviewer, we have made revisions to the suggested content in the manuscript, and this change has been marked in the text. Please refer to manuscript for detailed information (Line 126).

Round 2

Reviewer 3 Report

Comments and Suggestions for Authors

It is good to see that the authors have taken the comments to heart and changed the MS accordingly.

Comments on the Quality of English Language

Some minor edits of the English would increase the legibility and be advantageous to the reader., 

Author Response

Response: We thank the editor for the encouraging comments, according to the constructive comments of reviewer,  we have fully checked and polished the Englished writing.